# OpenReview forum: "Playing the Fool: Jailbreaking Large Language Models with Out-of-Distribution Strategies"
_ICLR.cc/2025/Conference — ICLR 2025 Conference Withdrawn Submission_

### Official Review · Reviewer_A6YU · 2024-10-27

**Soundness:** 3
**Presentation:** 3
**Contribution:** 3
**Rating:** 5
**Confidence:** 3

**Summary:**

The paper investigates the vulnerabilities in safety-aligned LLMs and MLLMs when exposed to out-of-distribution (OOD) inputs. Despite advancements in safety alignment, LLMs and MLLMs can still be "jailbroken" or manipulated into producing unsafe outputs when presented with novel, OOD inputs that differ from their training distribution. The authors propose a new jailbreak strategy, JOOD, which uses simple transformations—such as mixing images and text in novel ways—to bypass safety mechanisms by increasing the model's uncertainty in recognizing harmful content. This research highlights the need for improved safety-alignment mechanisms that can handle OOD inputs, as current methods are insufficient to prevent bypasses through subtle input manipulations.

Key contributions:

(1) The authors systematically analyze and demonstrate how RLHF (Reinforcement Learning from Human Feedback) safety-aligned models remain vulnerable to OOD inputs that do not fit within their trained safety-aligned data distribution.

(2) Proposing the JOOD, which leverages off-the-shelf transformation techniques, including text and image mixing, to create OOD-ifying inputs. These inputs can bypass safety protocols, effectively jailbreaking LLMs and MLLMs, including models like GPT-4 and GPT-4V.

(3) The study provides experimental evidence showing JOOD’s effectiveness across various jailbreak scenarios, achieving high attack success rates against state-of-the-art proprietary models, demonstrating that even simple transformations can lead to significant safety risks.

**Strengths:**

Originality:

While previous research has focused on jailbreaking through in-distribution adversarial prompts, this work introduces a novel perspective by highlighting how even simple OOD transformations can bypass safety mechanisms. The proposed JOOD strategy, which leverages straightforward techniques like text-mixing and image-mixing, is a creative way to demonstrate these vulnerabilities. This focus on OOD-ifying inputs marks a significant shift from existing approaches, providing a new understanding of the limitations in current safety-alignment strategies.

Quality:

The authors provide a comprehensive set of experiments across various scenarios that effectively validate the proposed JOOD method. The results are presented with clear metrics, such as attack success rate (ASR) and harmfulness scores, and show substantial improvements over baseline attack methods. Additionally, the authors conduct thorough ablation studies to analyze the impact of different transformation techniques, auxiliary images, and hyperparameters, ensuring the robustness and generality of their findings.

Clarity:

The use of diagrams and examples, such as Figure 1, helps to illustrate complex ideas and makes the methodology more accessible.

Significance:

The significance of this work lies in its ability to reveal critical weaknesses in current safety-alignment mechanisms for LLMs and MLLMs. By demonstrating that even simple OOD transformations can successfully bypass safety guardrails, the paper highlights a crucial area for future research and development. The findings are particularly important given the growing deployment of these models in real-world applications, where safety and ethical considerations are paramount. JOOD's demonstrated effectiveness against proprietary models like GPT-4 and GPT-4V underscores the urgency for more robust safety strategies. This work opens up new directions for the research community to explore better ways to handle OOD inputs and improve the resilience of safety mechanisms, making it a valuable contribution to the field.

**Weaknesses:**

1. The text-mixing transformations can lead to ambiguity where the blended words (e.g., "baopmpble") may not clearly combine into the intended original words. This can cause LLMs to interpret them in unexpected ways, leading to unpredictable outputs. A more detailed analysis of how consistent and reliable these transformations are in preserving the original intent would help clarify the robustness of the attack method.

2. The paper lacks sufficient comparisons with other state-of-the-art models like GPT-4o. Including benchmarks against a wider range of models would better highlight the strengths and limitations of the JOOD method and provide a clearer picture of its effectiveness.

3. Lack of Analysis on Text-Mixing Variants The paper introduces various text-mixing techniques but does not thoroughly analyze their individual effectiveness. A more detailed comparison of these methods would help identify which transformations are most successful, guiding future improvements to the approach.

**Questions:**

1. How can the model be guided to interpret the mixed words as the intended concepts?

The current text-mixing approach (e.g., creating words like "baopmpble") may lead to inconsistent interpretations. Have you considered how to ensure the model reliably understands these blended words as the intended concepts?

2. How can the model be made to generate actual content rather than explaining how to decode the mixed word?

In practice, models might try to explain or spell out the mixed word instead of generating content based on its intended meaning. Have you explored methods to prompt the model to directly produce the desired output rather than interpreting the word?

3. Why did JOOD fail to jailbreak models like GPT-4o and even GPT-4V?

When attempting to replicate the attack using the successful examples on GPT-4o and GPT-4V, the method was not consistently successful. Could you explain why this might be happening? What improvements might be needed for JOOD to be effective against more robustly aligned models?

---

### Official Review · Reviewer_qARz · 2024-10-31

**Soundness:** 2
**Presentation:** 3
**Contribution:** 2
**Rating:** 5
**Confidence:** 5

**Summary:**

The paper explores the vulnerability of large language models (LLMs) and multimodal large language models (MLLMs) when facing out-of-distribution (OOD) inputs. Despite these models being safety-aligned through human feedback and capable of generating safe, ethical, and fair responses in the presence of harmful inputs, they still face the risk of being "jailbroken" when handling OOD inputs — i.e., being prompted to generate outputs that violate safety and ethical standards.

To this end, the paper proposes a new "jailbreaking" strategy called JOOD, which is simple yet effective, utilizing text and image transformation techniques to generate OOD inputs, thereby increasing the model's uncertainty and effectively circumventing its safety measures. Experimental results show that even safety-aligned LLMs and MLLMs may still be "jailbroken" by simple OOD strategies.

**Strengths:**

1. The paper raises an important research question regarding the vulnerability of safety-aligned LLMs and MLLMs when facing OOD inputs. This is a significant concern, as existing models may pose safety risks when handling these OOD inputs.
2. The authors propose the JOOD strategy, which generates OOD inputs through simple text and image transformation techniques to bypass the model's safety measures. The method is straightforward and highly effective, with experimental results significantly surpassing existing approaches. This indicates that current LLMs and MLLMs are quite vulnerable to jailbreak attacks, which has important implications for advancing the development of safer LLMs and MLLMs.
3. The paper has a well-structured format and clear logic.

**Weaknesses:**

1. The description of the attack methods is not clear enough.
2. The experimental design is not comprehensive enough.

**Questions:**

1. My biggest concern is the novelty of the proposed technique. Could the authors provide a detailed discussion of the differences between JOOD and [R1,R2].
2. The JOOD attack strategy is very simple and effective. Are there any good defensive measures?
3. Regarding the attack methods, formulas (1), (2), and (3) do not include variables for text-mixing techniques and image-mixing techniques. During the experimental process, is it correct that all text mixing techniques (H-Interleave, V-Interleave, etc.) are applied to each text, and all image mixing techniques (RandAug, Mixup, etc.) are applied to each image? This means generating m×5 transformed instructions for each text and n×m×4 transformed instructions for each image, selecting the highest score among them. The reviewer would appreciate it if the setup could be clarified further.
4. Are Tables 2 and Figure 2 showing the attack effects specifically against GPT-4V? It is recommended to add a description in the table caption to indicate that it is a comparison of attack effects against GPT-4V. Why was the comparison only made for the MLLMs (GPT-4V)? Why is there no comparison regarding the LLMs (GPT-4)?
5. The attack effects of JOOD against LLMs should include a broader range of targeted models, such as open-source LLMs (llama, mistral, etc), not just limited to GPT-4.
6. For GPT-4, the attack success rate of JOOD on Social Violence is 0, which means that all text-mixing techniques have failed. What could be the possible reasons for this? Why did it not bypass the model's safety measures for this type of input?

Reference
- [R1] Erfan Shayegani, Yue Dong, and Nael Abu-Ghazaleh. Jailbreak in pieces: Compositional adversarial attacks on multi-modal language models. In The Twelfth International Conference on Learning Representations, 2023.
- [R2] Yichen Gong, Delong Ran, Jinyuan Liu, Conglei Wang, Tianshuo Cong, Anyu Wang, Sisi Duan, and Xiaoyun Wang. Figstep: Jailbreaking large vision-language models via typographic visual prompts. arXiv preprint arXiv:2311.05608, 2023.

---

### Official Review · Reviewer_SPzk · 2024-10-31

**Soundness:** 3
**Presentation:** 3
**Contribution:** 2
**Rating:** 3
**Confidence:** 4

**Summary:**

This paper designs a new jailbreak attack targeting LLMs and VLMs. This attack method uses text and image mixing techniques to transform input prompts into out-of-distribution data, thereby bypassing the safety alignment defenses of VLM models. The authors conducted some experiments, and the results show that OOD data is more likely to successfully jailbreak.

**Strengths:**

- The paper is well-written and easy to follow.
- The attack is quite effective compared to baselines.
- Discovering that OOD data is more susceptible to successful attacks is an interesting finding.

**Weaknesses:**

- Although the authors’ method of using OOD data for attacks is more effective than the baseline, the data mixing approach to obtain OOD data lacks significant innovation. Apart from this, the paper does not offer any other theoretical or technical innovations.
- While the authors proposed several different text or image mixing strategies, they did not introduce any technical or theoretical advancements.
- Such OOD data can be easily detected and filtered out. For example, the fluency of the prompt can be assessed in text, which is likely to reduce the effectiveness of the attack.
- Regarding word mixing, although the authors designed several strategies, there is no systematic method. For a new word and victim model, the attacker needs to carefully consider the strategy again and cannot immediately obtain an effective attack prompt.
- Although the authors conducted ablation studies, some issues remain unresolved, such as different tokenizers, which are very important for OOD words.
- Minor: Typo in “For OOD-ifying T^h into a novel text instruction, …”

Overall, this work is an interesting finding, but I think the contribution of this work is not sufficient to be accepted by ICLR.

**Questions:**

- “baopmpble” is an OOD word; how is it recognized by the tokenizer? OOD should be mapped to UNKNOWN tokens. Why does the model still understand it as meaning “bomb”?
- The authors claim in the article that mixed text like “baopmpble” can attack multiple black-box models. However, different models use different tokenizers. How does the same [WORD] achieve similar attack effects?
- If the malicious concept consists of multiple words rather than a simple word like “bomb,” how should it be mixed?

**Details Of Ethics Concerns:**

Ethical discussion should be included since the paper focuses on the jailbreak to LLM and VLM models.

---

### Official Review · Reviewer_djFe · 2024-11-11

**Soundness:** 3
**Presentation:** 3
**Contribution:** 3
**Rating:** 6
**Confidence:** 3

**Summary:**

This paper introduces JOOD, a novel jailbreak attack strategy that leverages out-of-distribution transformations to bypass the safety alignment of LLMs and MLLMs. By generating transformed inputs through techniques like Mixup, CutMix, and text-mixing technique variants (e.g., Interleaved and Concatenated text-mixing), the proposed approach effectively conducts the jailbreak attacks. The empirical evaluations across diverse attack scenarios demonstrate JOOD’s superior success rate compared to existing methods, demonstrating its capability to attack the open and closed source models, including GPT-4 and LLaVA.

**Strengths:**

1. This paper introduces an interesting angle to conduct jailbreak attacks from the out-of-distribution perspective.
2. The writing is easy to follow.

**Weaknesses:**

1. This paper proposes that multiple transformation techniques can be applied to generate the candidate inputs for jailbreak attacks. I am not clear on how many candidate inputs are necessary to conduct the attacks. Is there a strategy to guide the optimized input generation?

2. The evaluation seems not practical to me. From my understanding, it relies on external models to judge harmfulness, raising questions about the accuracy and reliability of these models as evaluators. I am concerned that the significant results reported may stem from biases or limitations within these external models rather than the actual effectiveness of the proposed attack. For instance, responses that are unrelated to the original harmful intent—such as teaching children survival information in case of a bomb threat—could potentially be flagged as harmful simply due to the presence of certain keywords. This could lead to false positives, where benign content is mistakenly categorized as harmful.

**Questions:**

See weakness.

---

### Note · Authors · 2024-11-15

**Comment:**

Dear Program Committee and Reviewers,

After careful consideration, we have decided to withdraw our paper from the conference review process. We would like to express our gratitude for the constructive feedback, which has provided valuable insights into how we can enhance our research. We look forward to carefully implementing these suggestions and potentially resubmitting an improved version to a future venue.

Thank you once again for your time and consideration.

**Withdrawal Confirmation:**

I have read and agree with the venue's withdrawal policy on behalf of myself and my co-authors.